# Theory of topological superconductivity and antiferromagnetic correlated insulators in twisted bilayer WSe$_2$

Chuyi Tuo[1,4], Ming-Rui Li[1,2,4], Zhengzhi Wu[1,3], Wen Sun[1] & Hong Yao[1] ✉

Since the very recent discovery of unconventional superconductivity in twisted WSe$_2$ homobilayers at filling $v = -1$, considerable interest has arisen in revealing its mechanism. In this paper, we developed a three-band tight-binding model with non-trivial band topology by direct Wannierization of the low-energy continuum model. Incorporating both onsite Hubbard repulsion and next-nearest-neighbor attraction, we then performed a mean-field analysis of the microscopic model and obtained a phase diagram qualitatively consistent with the experiment results. For zero or weak displacement field, the ground state is a Chern number $C = \pm 2$ topological superconductor in the Altland-Zirnbauer A-class (breaking time-reversal but preserving total $S_z$ symmetry) with inter-valley pairing dominant in $d_{xy} \pm id_{x^2-y^2}$–wave (mixing with a subdominant $p_x \mp ip_y$-wave) component. For a relatively strong displacement field, the ground state is a correlated insulator with the 120° antiferromagnetic order. Our results provide new insights into the nature of the twisted WSe$_2$ systems and suggest the need for further theoretical and experimental explorations.

In recent years, twisted van der Waals moiré superlattices[1–5] have garnered significant interest following the groundbreaking discovery of unconventional superconductivity (SC)[6–9] in magic-angle twisted bilayer graphene[10], which inspired numerous theoretical investigations on its microscopic origin[11–29]. Beyond graphene-based systems[30–43], twisted transition metal dichalcogenides (TMDs)[44,45] are regarded as a promising alternative platform to investigate many-body physics. Due to their high tunability, twisted TMDs can host a wide variety of exotic phases, including correlated insulators[46–52], integer and fractional quantum anomalous Hall states[53–57], and also integer and fractional quantum spin Hall states[58]. However, the experimental realization of SC in twisted TMD systems has remained elusive[48], though many theoretical works have suggested that SC should develop in such systems[59–69], typically through doping the correlated insulators. Very recently, two independent studies have reported the discovery of robust SC phases in twisted WSe$_2$ (tWSe$_2$) homobilayers[70,71]. The phenomenology of these SC phases differ significantly from that would

arise in doped Mott insulators, which calls for further theoretical investigations, particularly regarding the possible pairing mechanism and topological properties of SC, as well as the nature of adjacent correlated insulators.

Previous theoretical studies[48,59–66,72–75] of tWSe$_2$ homobilayers are largely based on the tight-binding description of the single-band moiré Hubbard model with spin-dependent hopping phase tuned by the displacement field. By formulating the problem in real space, this approach captures the locality of screened interactions and facilitates a clearer real-space insights essential for understanding the correlated physics. However, such a simple single-band model cannot capture the potential nontrivial band topology[45,76–79] and only applies to limited twist angles[48]. Thus, a direct Wannierization[79–85] of the standard continuum model[45] with a proper set of low-energy bands appears to be a more suitable approach to start with.

In this paper, we focus mainly on the experimental results in ref. 70, where the SC phase is observed in a 3.65° tWSe$_2$ device at

¹Institute for Advanced Study, Tsinghua University, Beijing, China. ²Department of Physics, Princeton University, Princeton, NJ, USA. ³Rudolf Peierls Centre for Theoretical Physics, Oxford, UK. ⁴These authors contributed equally: Chuyi Tuo, Ming-Rui Li. ✉e-mail: yaohong@tsinghua.edu.cn

integer filling factor $v = -1$ under a small displacement field, along with a correlated insulator phase in a larger displacement field. Using the continuum model parameters provided in ref. 79, we first construct a three-band tight-binding model for 3.65° tWSe$_2$ through direct Wannierization[79–85]. In addition to the onsite Hubbard repulsion, our model includes the next-nearest-neighbor (NNN) attraction, which can arise effectively through electron-phonon coupling[11–24,86,87] or purely electronic mechanisms[88–95]. By keeping these key interaction terms, our model can capture the essential qualitative physics of the experiment[70] and offer valuable real-space insights into the twisted WSe$_2$ system. We then perform a thorough mean-field analysis to the interacting model and obtain the phase diagram at filling $v = -1$ under realistic interaction strengths, concluding that the SC phase is consistent with inter-valley pairing with mixed $d_{xy} \pm i d_{x^2-y^2}$ and $p_x \mp i p_y$-wave symmetry, featuring nontrivial topology under Altland-Zirnbauer A-class[96–98] with Chern number $C = \pm 2$, and the correlated insulator phase has 120° antiferromagnetic (AFM) order. Additionally, the mixed symmetry character of the SC phase near zero displacement field predicted by our model has experiment observable distinctions from that of moiré Hubbard model[48,59–66,72–75,99,100] due to the absence of emergent spin-valley SU(2) symmetry[72,73], providing new insights into the nature of the moiré TMD systems.

## Results

### Continuum model and symmetry analysis

We begin with the standard low-energy continuum model description of tWSe$_2$[45]. The non-interacting continuum Hamiltonian for the $K$ valley is given by:

$$H_K(\boldsymbol{r}) = \begin{pmatrix} -\frac{\hbar^2(\boldsymbol{k}-\boldsymbol{\kappa}_+)^2}{2m^*} + \Delta_+(\boldsymbol{r}) & \Delta_T(\boldsymbol{r}) \\ \Delta_T^\dagger(\boldsymbol{r}) & -\frac{\hbar^2(\boldsymbol{k}-\boldsymbol{\kappa}_-)^2}{2m^*} + \Delta_-(\boldsymbol{r}) \end{pmatrix} \quad (1)$$

where $m^*$ is the effective mass of valence band edge, $\boldsymbol{\kappa}_\pm$ are located at corners of the mini Brillouin zone as shown in Fig. 1a. In the lowest-order harmonic approximation, we only keep terms with moiré wave vectors $\boldsymbol{g}_j$ which are obtained by rotation of $\boldsymbol{g}_1 = (\frac{4\pi}{\sqrt{3}a_M}, 0)$ by $(j-1)\pi/3$ in moiré potential $\Delta_\pm(\boldsymbol{r})$ and interlayer tunneling $\Delta_T(\boldsymbol{r})$:

$$\Delta_\pm(\boldsymbol{r}) = \pm \frac{V_z}{2} + 2V \sum_{j=1,3,5} \cos\left(\boldsymbol{g}_j \cdot \boldsymbol{r} \pm \psi\right) \quad (2)$$

$$\Delta_T(\boldsymbol{r}) = w\left(1 + e^{-i\boldsymbol{g}_2 \cdot \boldsymbol{r}} + e^{-i\boldsymbol{g}_3 \cdot \boldsymbol{r}}\right) \quad (3)$$

and the Hamiltonian for $-K$ valley is related by time-reversal symmetry $\mathcal{T}$. Throughout this paper, we adopt the experimental relevant twist angle $\theta = 3.65°$[70], continuum model parameters derived by large scale ab initio simulations $(V, \psi, w) = (9 \text{ meV}, 128°, 18 \text{ meV})$[79] and the effective mass $m^* = 0.45m_e$[101]. The moiré lattice constant is given by $a_M \approx a_0/\theta$ with $a_0 = 3.317 \text{ Å}$[102].

In the absence of displacement field $V_z = 0$, tWSe$_2$ system has $D_3$ point group symmetry generated by threefold rotation $C_{3z}$ and twofold rotation $C_{2y}$. Additionally, due to the lowest order harmonic approximation we adopted, the continuum model has additional pseudo-inversion symmetry $\mathcal{I}$ with $\sigma_x H_K(\boldsymbol{r})\sigma_x = H_K(-\boldsymbol{r})$, enlarging the point group symmetry to $D_{3d}$. When a finite displacement field $V_z \neq 0$ is applied, the point group symmetry is reduced from $D_{3d}$ to $C_{3v}$, breaking all symmetries that interchange the two layers. Apart from point group symmetries, the system also exhibits U(1) spin-valley symmetry and time-reversal symmetry $\mathcal{T}$ for both $V_z = 0$ and $V_z \neq 0$. These symmetry considerations are crucial for the subsequent construction of tight binding model and the classification of SC pairing symmetries.

### Wannier functions and tight-binding model

To study the low-energy physics, we first focus on the top moiré valence bands of the tWSe$_2$ continuum model at $V_z = 0$. The Chern numbers for each valley share the same sign up to top five moiré bands, precluding any low-energy real-space description with correct band topology due to Wannier obstructions. Therefore, we focus on accurately reproducing the Chern numbers of the top two moiré bands, and construct a minimal three-band tight-binding model with Chern numbers $(1, 1, -2)$ for $K$ valley. Following refs. 79–82, we construct sufficiently localized Wannier functions using their layer polarization properties and the location of Wannier centers. The resulting Wannier functions are shown in Fig. 1b, where the A and B orbitals are centered at XM and MX regions $(\pm a_M/\sqrt{3}, 0)$ with opposite layer polarization, and the C orbital is centered at MM region $(0, 0)$ with layer hybridization. Detailed symmetry analysis[81] reveals that the (A, B, C) orbitals for the $K$ valley have $C_{3z}$ eigenvalues $(e^{-i2\pi/3}, e^{i2\pi/3}, 1)$, and $C_{2y}\mathcal{T}$ or $\mathcal{I}$ symmetries interchange A and B orbitals but leave C orbital invariant. Using these Wannier functions, we can construct the following tight-binding model:

$$H_0 = \sum_{i\alpha j\beta\sigma} t_{i\alpha j\beta\sigma} c_{i\alpha\sigma}^\dagger c_{j\beta\sigma} \quad (4)$$

where $i, j$ are unit cell indexes, $\alpha, \beta$ are sublattice indexes (A, B, C), and $\sigma$ is spin-valley index $\uparrow$ (or $K$), $\downarrow$ (or $-K$). By keeping up to fifth-nearest-neighbor hopping parameters (see Supplementary Section I), Fig. 1c illustrates the close matching between the band structure of the tight-binding model and the continuum model, especially for the top moiré valence band.

We then consider the case of $V_z \neq 0$. Instead of repeating the above procedure for each $V_z$ separately, we utilize the layer polarization properties of Wannier functions, modeling the effect of displacement field as:

$$H_D = \frac{V_z}{2} \sum_i (n_{iA} - n_{iB}) \quad (5)$$

where $V_z$ is the energy difference between A and B sublattices induced by the displacement field, and $n_{i\alpha} = \sum_\sigma c_{i\alpha\sigma}^\dagger c_{i\alpha\sigma}$ is the density operator. Since the energy expectation value of displacement field on the C sublattice can be chosen as 0, we neglect terms involving it in $H_D$ (see Supplementary Section II for a more detailed discussion of the displacement field effects). The Fermi surfaces of filling factor $v = -1$ with $V_z = 0, 5, 15$ and 25 meV are shown in Fig. 2a, where the Fermi surface of spin up and down are split by the displacement field $V_z$, but related by the time-reversal symmetry $\mathcal{T}$. Figure 2b also illustrates the Fermi surface density of states (DOS) as a function of $V_z$.

To capture the many-body physics, we consider leading interactions in tWSe$_2$. The dominant interaction should be the onsite Hubbard repulsion:

$$H_U = \sum_{i\alpha} U_\alpha n_{i\alpha\uparrow} n_{i\alpha\downarrow} \quad (6)$$

where $\alpha = A, B, C$ labels the sublattice, and $n_{i\alpha\sigma} = c_{i\alpha\sigma}^\dagger c_{i\alpha\sigma}$. The positive Hubbard interaction can typically lead to magnetic ordering, providing a promising explanation of the correlated insulator phase. Longer-range Coulomb repulsions are neglected mainly because their magnitudes are much smaller (see Supplementary Section IV).

Due to the time-reversal symmetry $\mathcal{T}$ of the systems, the Fermi surface features Cooper instability; namely, SC instabilities can occur even with infinitesimal attractions. Considering that the

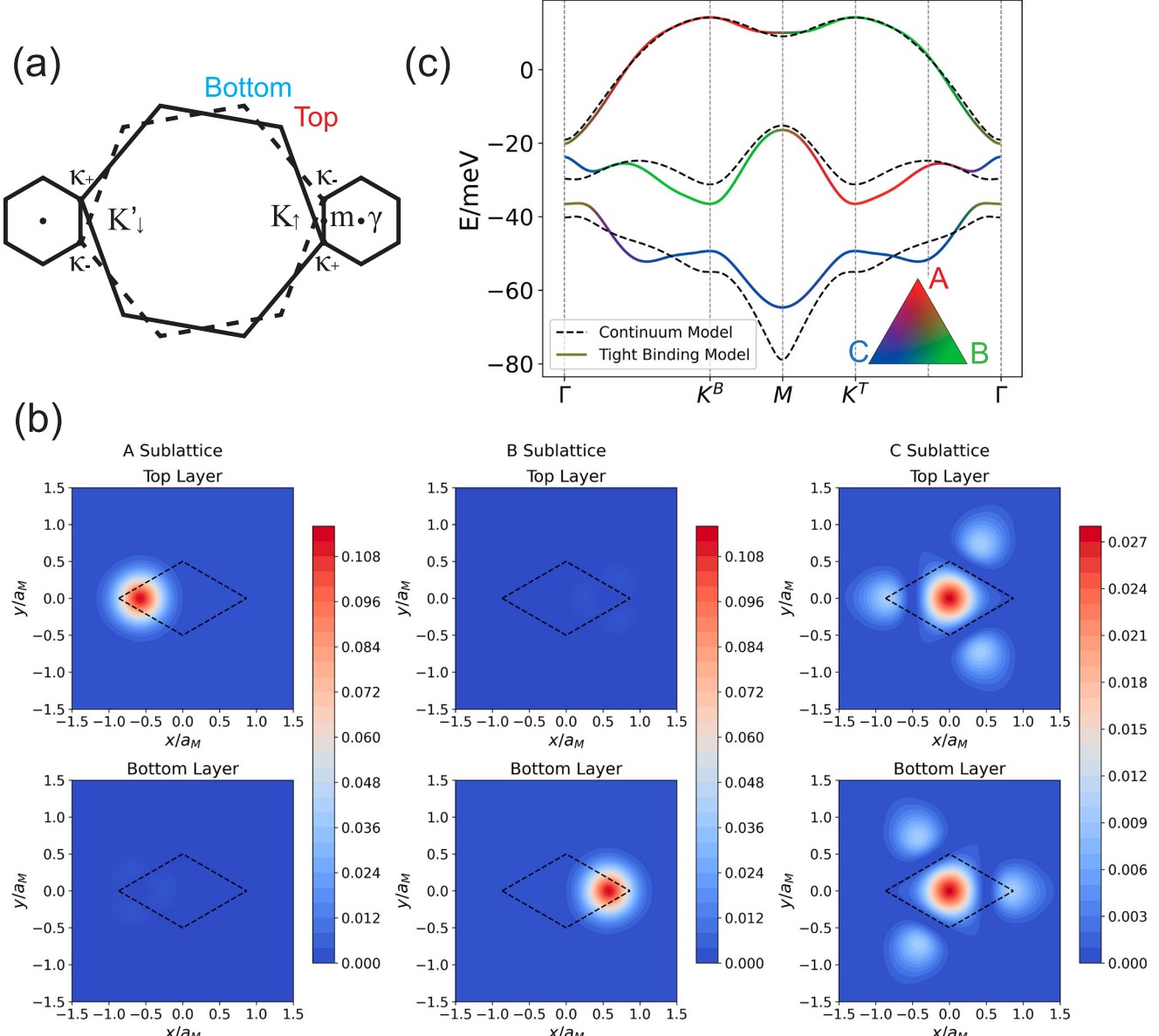

Fig. 1 | Construction of the three-band tight-binding model from the continuum model by Wannierization. a Mini Brillouin zone formed by a small twist angle $\theta$ between two layers. b Density distribution of Wannier functions on two layers, with unit cell shown as dashed line. c Comparison of the band structure between continuum model (dashed line) and tight binding model (solid line), where colors representing sublattice components.

Fermi surface shown in Fig. 2a favor pairing within the same sublattices and has almost no C-orbital component, a natural choice is to consider the attractions on NNN sites of the same A or B sublattices:

$$H_{V_2} = -V_2 \sum_i \sum_{\alpha \in \{A,B\}} \sum_{\delta \in \text{NNN}} n_{i+\delta\alpha} n_{i\alpha} \quad (7)$$

where $V_2$ is the strength of NNN attraction, $\alpha$ is sublattice index of A and B, and $\delta$ represents one of the three NNN bond directions 120° apart. Besides, we have also considered the effects of nearest-neighbor (NN) attraction between A and B sublattices in Supplementary Section VI, where we have shown that it is less dominant. The relatively local NNN (or NN) attraction can be understood by a random phase approximation (RPA) analysis (see Supplementary Section V). Therefore, the interacting model which we should consider to describe the main physics of tWSe$_2$ is denoted as $H = H_0 + H_D + H_U + H_{V_2}$.

## Mean-field analysis of SC and AFM

We start with the mean-field analysis of SC instabilities by decoupling the NNN attraction $H_{V_2}$ in the SC channel as:

$$H_{V_2} \approx -V_2 \sum_{i\sigma\sigma'} \sum_{\alpha \in \{A,B\}} \sum_{\delta \in NNN} \left( \tilde{\Delta}_{\alpha\sigma\sigma'}^*(\delta) c_{i\alpha\sigma'} c_{i+\delta\alpha\sigma} \right. \\ \left. + c_{i+\delta\alpha\sigma}^\dagger c_{i\alpha\sigma'}^\dagger \tilde{\Delta}_{\alpha\sigma\sigma'}(\delta) - \tilde{\Delta}_{\alpha\sigma\sigma'}^*(\delta) \tilde{\Delta}_{\alpha\sigma\sigma'}(\delta) \right), \quad (8)$$

where $\tilde{\Delta}_{\alpha\sigma\sigma'}(\delta) = \langle c_{i\alpha\sigma'} c_{i+\delta\alpha\sigma} \rangle$ is the spatially uniform real-space pairing order parameter on NNN bond $\delta$, as shown in Fig. 3a. To further determine the ansatz of the pairing order parameter $\tilde{\Delta}_{\alpha\sigma\sigma'}(\delta)$, we analyze possible pairing symmetries, which should be classified by the irreducible representations of the symmetry group. Since the addition of displacement field $\mathcal{V}_z$ preserves time-reversal symmetry $\mathcal{T}$ but breaks pseudo-inversion symmetry $\mathcal{I}$, we only consider the $S_z = 0$ sector for the U(1) spin-valley symmetry (i.e. inter-valley pairing). And these $S_z = 0$ pairings can be further categorized into singlet

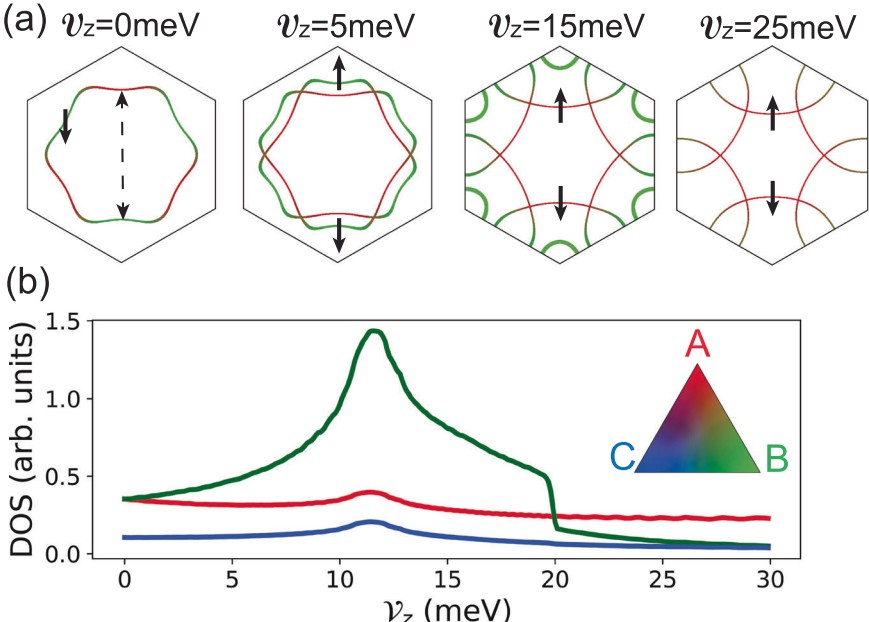

**Fig. 2 | Evolution of Fermi surfaces and density of states as a function of displacement field. a** Fermi surfaces of free Hamiltonian at different displacement fields, with colors indicating the sublattice components, and thickness representing the DOS. The spin of the Fermi surfaces are labeled as solid arrows. When $V_z = 0$ meV, where spin up and down Fermi surfaces coincide, only spin down Fermi surface is shown. The approximate nesting wave vector $\mathbf{Q}$ between spin up and down Fermi surfaces are illustrated as dashed arrow. **b** The Fermi surface DOS of A, B, C sublattices as a function of displacement field $V_z$.

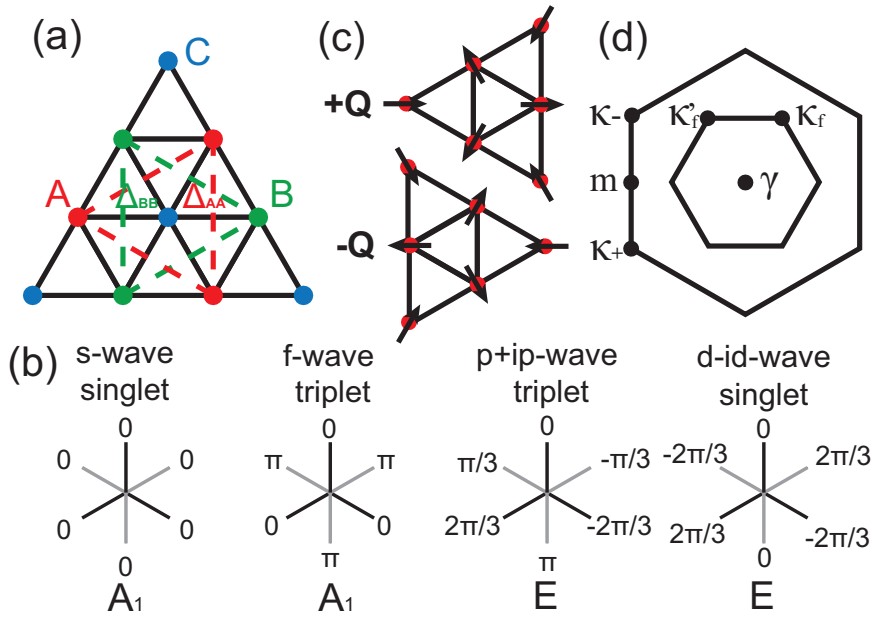

**Fig. 3 | Illustration of the possible superconducting and antiferromagnetic order parameters. a** The NNN SC order parameters $\Delta_{AA}$ and $\Delta_{BB}$, where the A, B, C sites are labeled as red, green, blue dots. **b** The pairing form factor of $s$-wave, $f$-wave, $p + ip$-wave, and $d - id$-wave on NNN bonds, with the irreducible representation labeled. **c** 120° AFM pattern on a certain type of sublattice with wave vector $\pm Q$. **d** The folded Brillouin zone induced by the AFM order.

pairing $\Delta_\alpha^S(\delta) = (\tilde{\Delta}_{\alpha\uparrow\downarrow}(\delta) - \tilde{\Delta}_{\alpha\downarrow\uparrow}(\delta))/\sqrt{2}$ and triplet pairing $\Delta_\alpha^T(\delta) = (\tilde{\Delta}_{\alpha\uparrow\downarrow}(\delta) + \tilde{\Delta}_{\alpha\downarrow\uparrow}(\delta))/\sqrt{2}$. For the point group symmetry $C_{3v}$ under finite displacement field, it is straightforward to show that only $A_1$ (mixing $s$ and $f$-wave) and $E$ (mixing $(p_x, p_y)$ and $(d_{xy}, d_{x^2-y^2})$–wave) representations are possible for NNN pairing. The SC pairing form factors, focusing only on chiral SC in the $E$ representation, are illustrated in Fig. 3b, and we leave the discussion of nematic SC in the $E$ representation in Supplementary Section VI, where we have shown it is subdominant.

To understand the competing insulating phase, we decouple the onsite Hubbard repulsion $H_U$ in the channel of magnetic ordering in the $xy$-plane[72,73] as:

$$H_U = -\sum_{i\alpha} U_\alpha \left( m_{i\alpha} c_{i\alpha\downarrow}^\dagger c_{i\alpha\uparrow} + m_{i\alpha}^* c_{i\alpha\uparrow}^\dagger c_{i\alpha\downarrow} \right)$$
$$+ \sum_{i\alpha} U_\alpha |m_{i\alpha}|^2 + \frac{1}{2}\sum_{i\alpha\sigma} U_\alpha n_{i\alpha\sigma} \qquad (9)$$

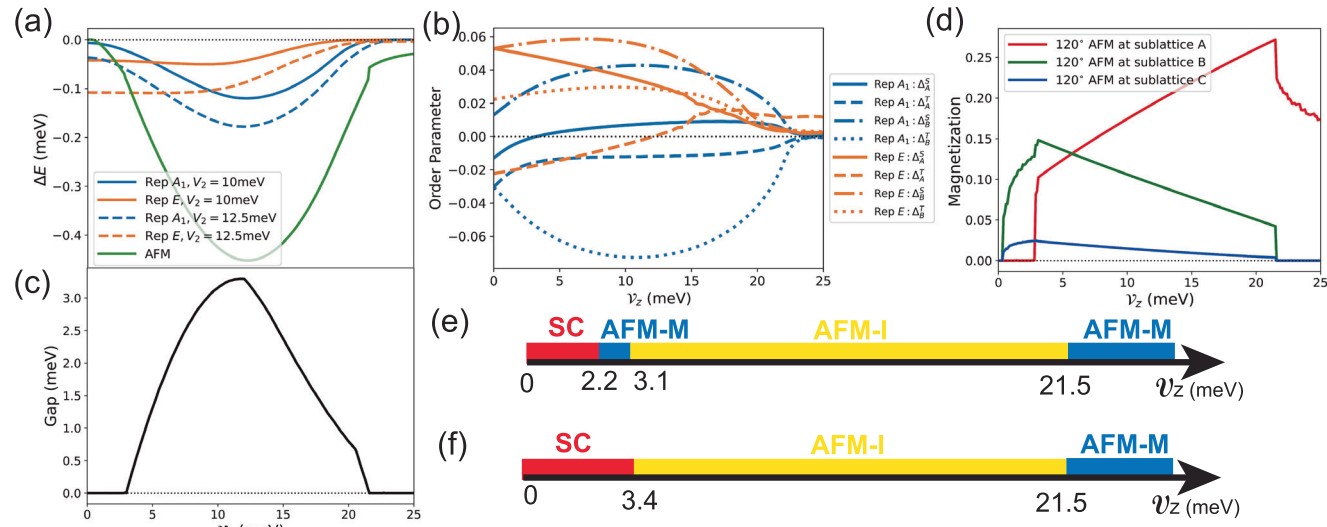

**Fig. 4 | Mean-field analysis and phase diagrams of 3.65° tWSe₂. a** The energy gain per hole of the ordered phases compared to the symmetric phase. **b** SC orders for $V_2 = 10$ meV on one specific NNN bond $\delta_0$ under the gauge described in the main text. The sign of order parameters correspond to the sign of real (imaginary) part of singlet (triplet) pairings. **c** The AFM mean-field charge gap at filling factor $\tilde{\nu} = -3$ in the folded Brillouin zone. **d** Magnitude of the AFM orders on A, B, C sublattices. **e**, **f** The phase diagrams for $V_2 = 10$ meV and $V_2 = 12.5$ meV respectively, indicating that under mean-field framework, the SC to AFM-I transition can either have an intermediate AFM-M phase or occur directly.

where $m_{i\alpha} = \langle c^{\dagger}_{i\alpha\uparrow} c_{i\alpha\downarrow} \rangle = \langle S^x_{i\alpha} \rangle + i \langle S^y_{i\alpha} \rangle$ is the complex order parameter representing the in-plane magnetization. We further constraint the form of $m_{i\alpha}$ by noting that, as shown in Fig. 2a, although the Fermi surface deforms as the displacement field $\mathcal{V}_z$ changes, an approximate nesting between spin up and spin down Fermi surfaces persist. Moreover, the nesting wave vectors are relatively close to the commensurate ones $\pm \mathbf{Q} = (0, \pm 4\pi/3a_M)$ over a wide range of displacement field $\mathcal{V}_z$, giving raise to the 120° AFM order $m_{i\alpha} = m^+_\alpha e^{i\mathbf{Q}\cdot\mathbf{R}_i} + m^-_\alpha e^{-i\mathbf{Q}\cdot\mathbf{R}_i}$ with two possible chiralities as shown in Fig. 3c, which are generally non-degenerate under finite $\mathcal{V}_z$. Such AFM order breaks translation symmetry of the system, folding the Brillouin zone as shown in Fig. 3d, with the new filling factor in terms of the folded Brillouin zone becoming $\tilde{\nu} = -3$. Depending on details of the system, the ground state can be either metallic or insulating due to the nesting of $\pm \mathbf{Q}$ is non-perfect.

We then perform mean-field calculations for SC and AFM at filling $\nu = -1$ independently, using Hubbard repulsion $U_A = U_B = U_C = 37.5$ meV and two different NNN attractions $V_2 = 10$ meV or $V_2 = 12.5$ meV as representing parameters. These interacting parameters can be estimated either by comparing the mean-field results with experimental observations, or by directly expanding the gate-screened Coulomb interaction onto Wannier functions (see Supplementary Sections III and IV). The resulting phase diagrams for tuning the displacement field $\mathcal{V}_z$ at filling $\nu = -1$ are summarized in Fig. 4e and f, respectively. We leave more detailed mean-field derivations and discussions in Supplementary Sections VI and VII.

**Topological superconductivity**

We first focus on the SC phase in the zero- or small-displacement field regime. A comparison of the energy gain $\Delta E$ between possible SC orders and the 120° AFM order is shown in Fig. 4a, which indicates that the system is in the SC phase for $\mathcal{V}_z < \mathcal{V}_{z,c} \approx 2.2$ meV (3.4 meV) for $V_2 = 10$ meV (12.5 meV), qualitatively consistent with the critical field $\mathcal{V}^{\text{exp}}_{z,c} \approx 2.6$ meV observed in the experiment[70]. The pairing of this SC phase is chiral with mixed $d_{xy} \pm i d_{x^2-y^2}$ and $p_x \mp i p_y$ wave in the E representation of $C_{3v}$, spontaneously breaking the time-reversal symmetry $\mathcal{T}$[103]. More importantly, such a chiral SC phase, which fits into the A-class (namely, breaking $\mathcal{T}$ but preserving the $S_z$) of the Altland-Zirnbauer tenfold classification

scheme[96–98], is topologically non-trivial with Chern number computed to be $C = \pm 2$, suggesting tWSe₂ as a promising candidate for realizing chiral topological SC. The internal structure of the SC order as a function of $\mathcal{V}_z$ for $V_2 = 10$ meV is also illustrated in Fig. 4b, where the consistency constraint between time-reversal symmetry $\mathcal{T}$ and $C_{3v}$ mirror plane symmetry enables us to fix a simple gauge such that all singlet pairings $\Delta^S_\alpha(\delta)$ are real while all triplet pairings $\Delta^T_\alpha(\delta)$ are imaginary. It is worth emphasizing that, unlike the single-band moiré Hubbard model[48,59–66,72–75,99,100], the mixing of singlet and triplet pairings is allowed for all $\mathcal{V}_z$ in our three-band model, especially for $\mathcal{V}_z = 0$, where the emergent spin-valley SU(2) symmetry is absent and the pseudo-inversion symmetry $\mathcal{I}$ does not forbid such mixing. Also, the dominance of the $d_{xy} + i d_{x^2-y^2}$ component over the $p_x \mp i p_y$ one in the SC phase is in accordance with the result of Chern number $\pm 2$. Moreover, SC is enhanced (suppressed) on the B (A) sublattice upon increasing the displacement field $\mathcal{V}_z$ from zero, closely following the trend of the Fermi surface DOS in Fig. 2b.

**Antiferromagnetic correlated insulators**

As the displacement field is further increased beyond a critical field $\mathcal{V}_{z,c}$, the system transitions into the 120° AFM phase. To further determine its transport properties, we compute the AFM mean-field charge gap for $\tilde{\nu} = -3$. As shown in Fig. 4c, a broad intermediate range of the AFM insulator (AFM-I) phase appears for $\mathcal{V}_{z,c} \lesssim \mathcal{V}_z < \mathcal{V}'_{z,c} \approx 21.5$ meV and an AFM metal (AFM-M) phase for $\mathcal{V}_z > \mathcal{V}'_{z,c}$, closely matching the phenomenology of the experiment[70]. In the mean-field framework, depending on the value of $V_2$ employed in the model, the SC to AFM-I transition either exhibits a tiny intermediate AFM-M phase as shown in Fig. 4e or occurs as a direct first-order transition as shown in Fig. 4f. To explain the transport evidence of continuous superconductor-insulator transition[70], disorder could play an important role. The continuous transition into the SC phase might potentially be a percolation transition[104,105] of local SC regions induced by disorders or a disorder-rounded first-order transition. We also present the magnitude of AFM orders on different sublattices as a function of $\mathcal{V}_z$ in Fig. 4d. Except for small $\mathcal{V}_z$, where the AFM order is stronger on the B sublattice due to its larger DOS (see Fig. 2b), the AFM order generally favors the A sublattice, as the holes are mostly concentrated there in

response to the displacement field $\mathcal{V}_z$, as well as its better approximation for the commensurate nesting wave vector $\boldsymbol{Q}$. And the sudden drop of AFM orders at large $\mathcal{V}_z$ regime coincides with the disappearance of B sublattice hole pockets at $\kappa_\pm$ points and the sharp decline in B sublattice DOS, as illustrated in Fig. 2, indicating that the holes on B sublattice play a crucial role in mediating the AFM order.

## Discussion

In summary, we have constructed a three-band tight-binding model through direct Wannierization, and incorporated onsite Hubbard repulsion and NNN attraction to explain the SC and correlated insulator phase observed in the 3.65° tWSe$_2$ at filling $v = -1$[70]. Our mean-field analysis indicates that, the SC phase is an A-class topological SC with Chern number $C = \pm 2$, featuring the inter-valley mixed $d_{xy} \pm id_{x^2-y^2}$ and $p_x \mp ip_y$-wave pairing symmetry, and the correlated insulator phase is explained by the 120° AFM order. We further demonstrate that the metallic behavior when the filling is away from $v = -1$ can also be qualitatively understood within our mean-field framework (see Supplementary Section VIII). Compared with the 5° tWSe$_2$ system[71], the experimental features show notable similarities despite differences in low-energy band structure and interaction strength[82]. Recent theoretical advances[106] suggest the possibility of a unifying underlying mechanism, motivating future efforts toward a unified description.

The topological band structure in our model can be crucial for understanding how SC arises for flat band systems, since non-trivial lower bounds of SC superfluid weight exist[107–112] due to quantum geometric effects, which deserves more detailed future theoretical studies. Moreover, our results also suggest that the tWSe$_2$ homobilayers could provide new possibilities for realizing topological SC, which call for more detailed experimental studies to further uncover its nature. Identifying the topological edge states by measuring quantized thermal Hall conductance[113–115], or verifying the chiral nature of the SC pairing symmetry through the phase-sensitive Josephson junction[116–118], will surely open new opportunities in twisted TMD systems.

### Note added

In finishing the present work, we noticed that refs. 119–124 appeared, which also investigated the SC and correlated insulators in tWSe$_2$ reported in ref. 70, although there are important differences between those studies and the present one.

## Methods

We construct a three-band tight-binding model by a direct Wannierization of the continuum model of 3.65° twisted bilayer WSe2, with the resulting hopping parameters given in Supplementary Information Section I. The model incorporates onsite Hubbard repulsion and nearest- or next-nearest-neighbor attractions. A detailed estimation of the Hubbard interaction parameters is provided in Supplementary Information Sections III and IV, while a microscopic explanation of the attractive terms is presented by the random phase approximation calculation in Supplementary Information Section V. The interacting model is analyzed within the standard mean-field framework, with full derivations and self-consistent procedures provided in Supplementary Information Sections VI and VII.

## Data availability

Source data for all figures in the main article are available in the Supplementary Data file. Source data are provided with this paper.

## Code availability

The codes used in this study are available from the corresponding author upon request.

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

## Acknowledgements

We would like to thank Andrei Bernevig for helpful discussions. This work is supported in part by MOSTC under Grant No. 2021YFA1400100 (H.Y.), by the Innovation Program for Quantum Science and Technology under Grant No.2021ZD0302502 (H.Y.), by NSFC under Grant Nos. 12347107 (C.T., M.-R.L., Z.W., W.S., H.Y.) and 12334003 (H.Y.), and by the New Cornerstone Science Foundation through the Xplorer Prize (H.Y.).

## Author contributions

H.Y. conceived and supervised the project. C.T. and M.-R.L. carried out the computations. All authors (C.T., M.-R.L., Z.W., W.S., and H.Y.) contributed to the interpretation of results and to writing the manuscript.

## Competing interests

The authors declare no competing interests.
