## [Transparent Peer Review file · Nature Communications]

Theory of Topological Superconductivity and Antiferromagnetic Correlated Insulators in Twisted Bilayer WSe₂

Corresponding Author: Professor Hong Yao

Version 0:

Reviewer comments:

Reviewer #1

(Remarks to the Author)

The authors aimed to provide a mean-field explanation of the recent experimentally observed phase diagram of twisted bilayer WSe₂ (3.65 degrees). The many-body Hamiltonian contains a 3-band (per spin) Wannier model that properly captures the first two bands per spin, an onsite repulsion, and an NNN attraction. With mean-field approximation, the authors obtained a chiral topological superconductor phase at a small displacement field, an AFM insulator at an intermediate displacement field, and an AFM metal at a large magnetic field for hole filling being 1.

The importance of this work relies on its experimental relevance. The obtained superconductor-insulator-metal behavior is consistent with the experiment, which is a huge plus. The authors further provided some predictions about the three phases--nontrivial topology for the superconducting phase and the AFM for the rest, which will certainly motivate future efforts. The only big issue that prevents me from directly recommending the publication of the current version is the reasoning behind the chosen interaction terms. I can imagine the onsite repulsion, but it is hard for me to imagine (and thus easily accept) the NNN attraction, as the authors did not provide any detailed explanation of such a term. The authors mentioned that the NN attraction is less dominant, but why we should choose ~10meV for the NNN attraction? What is the logic behind it?

So I recommend the authors provide more support for the chosen value of the NNN attraction. If the authors can address this request, I would be happy to recommend the publication. Also, it would be good to provide some support for the chosen value of U.

One small issue: the label for the NNN attraction is V in equations 7 and 8, and is changed to V₂ in figure 4. I suggest the authors make the label consistent by sticking to V₂, since V is already used in equation 2.

Reviewer #2

(Remarks to the Author)

In this work, after constructing a three-band tight-binding model through direct Wannierization and incorporating onsite Hubbard repulsion, the authors theoretically analyzed superconducting and correlated insulator phases within mean-field theory as a displacement field changes at a certain twist angle and filling factor, explaining recent experimental results by Y. Xia et al. in Ref. [71] on twisted bilayer WSe₂. The manuscript demonstrates that for zero or weak displacement fields, a topological superconducting phase with a nontrivial Chern number occurs, while for a relatively strong displacement field, the ground state becomes a correlated antiferromagnetic insulator. The authors argue that the theoretical results are largely consistent with the recent experiments in Ref. [71].

Since mean-field theory typically overestimates a tendency toward broken symmetry states and it is sometimes possible to find a parameter set that exhibits a desired broken symmetry, the proposed mean-field theory needs to be tested with a broader range of parameters for more convincing comparisons with experiments, with a discussion on the validity of the

theory. Here are some questions or comments that, in my opinion, need to be clarified in the revised version:

- The mean-field phase diagrams depend on the detailed choice of interaction parameters, such as the Hubbard repulsions U_A , U_B , and U_C , and next-nearest-neighbor (NNN) attractions V . How can the choice of parameters used in the manuscript be justified? Are the values of these parameters simply chosen to match the observed phases in Ref. [71], or are these parameters estimated independently?

- The authors only considered a filling $\nu=1$. What is the doping dependence of the superconducting phase? In Ref. [71], different metallic behaviors were observed above and below the filling $\nu=1$. Does the proposed mean-field theory reproduce this type of behavior consistent with the experiment?

- The authors only considered a twist angle of 3.65° . What would be the dependence on the twist angle? In Ref. [72], superconductivity was observed in twisted bilayer WSe₂ at a twist angle of 5° , where correlations may not play a significant role. Is the current theory still applicable to this case, and are the results consistent with the experiment in Ref. [72]? The authors should discuss the results of the experiment in Ref. [72], comparing them with the mean-field calculations.

Reviewer #3

(Remarks to the Author)

Version 1:

Reviewer comments:

Reviewer #1

(Remarks to the Author)

Unfortunately, I am not convinced by the authors' justification for the next-nearest-neighbor (NNN) attractive interaction. After reviewing their response, I find no independent reasoning for selecting this interaction other than numerically reproducing the experimentally observed phases. Given this, I have to recommend that the manuscript be published in a more specialized journal rather than Nature Communications.

(Remarks on code availability)

Reviewer #3

(Remarks to the Author)

The field of superconductivity in semiconductor moiré materials is at a very early stage, as there are only two experimental reports up to date. Besides, the model parameters that enter in the continuum descriptions of WSe₂ are obtained from DFT, which may have high uncertainties and depend on the functionals used. In this context, the approach taken from the authors to isolate the key physics rather than incorporating the full microscopic details is justified. The additional sections added to the supplemental material help supporting this point, as they show that the values taken for the interaction terms qualitatively reproduce several aspects of the experimental phenomenology. In the previous version of the manuscript the values taken for the mean-field Hamiltonian parameters were lacking a better justification. I recommend that the authors add a short comment in the introduction about the approach taken, something along the lines of what they comment in the reply: that they expect their choices to capture the essential qualitative physics of the system.

The authors have address all of my concerns. Because they are focusing on $\nu=1$, I agree that it is justified to just keep on-site interactions to capture the main physics and I also think it's okay to leave an explanation of the microscopic origin of attractive interactions for future work. I only have one final question: Have the authors numerically checked that the top band has no C-orbital component in the large V_z limit? How independent of V_z are the A and B Wannier functions really?

Regarding the reply to Reviewer 2: I think the authors provide satisfactory answers. I would suggest to add a comment in the discussion section about the similarities and differences of the 3.65° degree system that they study with respect to the one at 5° degree twist (mainly summarizing what they discuss on the reply.) I believe this would improve the quality of the paper.

For the previous reasons I recommend this work to be published in Nature Communications.

(Remarks on code availability)

Version 2:

Reviewer comments:

Reviewer #3

(Remarks to the Author)

The authors have satisfactorily addressed my additional question. I recommend this work to be published in Nature Communications.

(Remarks on code availability)

Reviewer #4

(Remarks to the Author)

Reviewer #1's primary concern about this manuscript pertained to the justification of the next-nearest-neighbor attractive interaction in the model. This attractive interaction is essential for the emergence of superconductivity, according to the authors' analysis.

In the resubmitted supplementary material, the authors present an RPA analysis for the renormalization of interactions. While this RPA analysis leads to a weaker attractive interaction strength compared to the values of V_2 adopted in the main text, it does demonstrate a microscopic mechanism for the appearance of an attractive interaction at a finite range in the twisted WSe₂ system. In my view, the reviewer #1's concern has been largely addressed.

(Remarks on code availability)

Response to Reviewers' Comments for Manuscript NCOMMS-24-59815

Chuyi Tuo,* Ming-Rui Li,* Zhengzhi Wu, Wen Sun, and Hong Yao†
Institute for Advanced Study, Tsinghua University, Beijing 100084, China

We sincerely thank all three reviewers for their insightful and constructive feedback, which has been invaluable in enhancing the quality and clarity of our manuscript. We have thoroughly revised the manuscript based on the reviewers' comments and suggestions.

The key revisions made to the manuscript are summarized as follows:

1. *Two additional sections are added to the Supplemental Material to further justify our choices of interacting parameters:*
“II. Mean-Field Results with Different Interacting Parameters”
“III. Interacting Parameters Directly Obtained from Wannier Functions”
2. *An additional section is added to the Supplemental Material to explain the metallic behavior away from filling $\nu = -1$:*
“VI. Mean-Field Results away from Filling $\nu = -1$ ”

Before diving into detailed point-to-point responses, we would like to first address the key concerns regarding our mean-field model and parameter choices below:

Concern 1: Model Terms

We first address the concern regarding why our mean-field model includes only onsite Hubbard repulsion and NNN (or NN) attraction as interaction terms.

In the literature, there are primarily two styles of mean-field studies. The first approach incorporates full microscopic interactions, while the second focuses on retaining only the terms most relevant to the main physics under study. Our mean-field approach follows the latter philosophy. In our view, these two approaches differ fundamentally in their objectives: the first seeks to precisely calculate physical properties, whereas the second aims to identify the key interactions based on experimental observations. As research on superconductivity in twisted WSe₂ is still in its early stages, our priority is to understand its mechanism rather than to achieve quantitative precision. Therefore, we believe that it is appropriate to retain only the most relevant terms in our model. And we conduct our mean-field study using a three-band tight-binding model, which capture the non-trivial band topology and provide valuable real space insights into twisted WSe₂ system.

For the mean-field study of the superconducting phase, it is necessary to phenomenologically include some attraction terms. Due to the time-reversal symmetry of twisted WSe₂, superconductivity emerges even with infinitesimal attraction since the Fermi surface Cooper instability occurs (see the Parameter Choices section below for the rationale behind choosing relatively large V_2 in the main text). By comparing the effects of NN (Supplemental Material) and NNN (main text) attractions, we identify NNN attraction as the key interaction for superconductivity, as it leads to significantly stronger superconducting behavior. This can be attributed to the strong sublattice polarization of the Fermi surface, where pairing between the same sublattice is favored. These phenomenologically included attractions may originate microscopically from electron-phonon coupling, fluctuations of neighboring insulating phases, or RPA-like overscreening effects.

For the mean-field study of the correlated insulating phase, it is crucial to identify an order in the particle-hole channel that fully gaps the Fermi surface. Our three-band model maintains approximate nesting between the spin-up and spin-down Fermi surfaces as the displacement field increases (see Fig. 2 in the main text), favoring $\sqrt{3} \times \sqrt{3}$ commensurate order for intermediate-to-strong coupling systems such as twisted WSe₂. The new filling factor

* These two authors contribute equally in this work

† yaohong@tsinghua.edu.cn

$\tilde{\nu} = -3$ in the enlarged unit cell indicates that the second band is half-filled. To explain the insulating behavior, the order must break the time-reversal symmetry to lift the spin-related degeneracy $E_{\uparrow}(k) = E_{\downarrow}(-k)$, naturally suggesting some magnetic order. We thus identify the onsite Hubbard repulsion as the key interaction for the correlated insulator due to its dominance in magnitude and tendency for magnetic order. We neglect longer-range interaction terms for two main reasons. First, the gate distance is comparable to the moire lattice constant (see Device fabrication section in [Nature 637, 833 (2025)]), leading to substantial screening effect. Second, longer-range repulsion (e.g., nearest-neighbor repulsion V_1) typically favors the charge density wave order without breaking the time-reversal symmetry, making it less relevant to the insulating behavior.

We now provide a brief comment on the explanation of the experimentally observed continuous superconductor-insulator transition. Within Landau symmetry-breaking paradigm, it is challenging to explain the continuous transition between superconducting phase and antiferromagnetic insulating phase because they break different symmetries (one exotic scenario is the deconfined quantum criticality, see [arXiv: 2406.12971] for further details). Here, we adopt a more conventional and plausible viewpoint, attributing the continuous transition behavior to disorder, which is notably present in twisted TMD systems. Disorder can induce local superconducting regions near the transition point, and the continuous transition observed in the transport experiment can naturally be explained by a percolation transition of these local regions.

Therefore, we conclude that our three-band tight-binding model with onsite Hubbard repulsion and NNN (or NN) attraction should be suitable (at least qualitatively) for describing the physical properties of the twisted WSe₂ system at filling $\nu = -1$.

Concern 2: Parameter Choices

We then address the concern about the parameter choices ($U_A = U_B = U_C = 37.5$ meV, $V_2 = 10$ meV or 12.5 meV) for our mean-field calculations.

Before discussing the interactions, we first review and comment on our non-interacting parameter choice. At this stage, the continuum model band parameters for moire TMD systems are primarily determined by large-scale DFT simulations on certain commensurate twist angles due to the experimental challenges of spectroscopy measurements. These parameters can depend sensitively on the DFT exchange-correlation functional for van der Waals interactions, and are only reliable near the DFT-simulated twist angle due to moire lattice relaxation effects. To the best of our knowledge, at the time of our research, the only available continuum model parameters for twisted WSe₂ were provided by Liang Fu's group in an early study [Nature Communications 12, 6730 (2021)], based on DFT simulations at a relatively large twist angle of 5.08°. Our tight-binding model is built on this set of continuum model parameters, which we believe qualitatively captures the 3.65° twisted WSe₂ system. However, quantitative discrepancies are expected and we anticipate future studies will improve the accuracy.

We now turn to the discussion on our interacting parameter choices. Considering the greater difficulty in accurately determining interaction strengths (due to the uncertainty of dielectric constant, screening length, and the screening effects from high-energy bands) and our emphasis on qualitative physics, we treat all interacting parameters as free and gain physical insights by adjusting these parameters and comparing the mean-field results with experiments. Since our model only includes onsite Hubbard repulsion and NNN attraction (NN attraction can be discussed similarly), there appear to be four free parameters: $U_A = U_B$, U_C , $V_{2,A} = V_{2,B}$, $V_{2,C}$. However, due to the C orbital is far from the Fermi surface for $\nu = -1$ (see Fig.2 in the main text), we expect it to be irrelevant to the main physics (more evidence for this can be found in the Supplemental Material Section II). Therefore, as in the main text, we can set $U_C = U_A = U_B$ and $V_{2,C} = 0$ without affecting the physical results. This simplification leaves only two physically relevant tuning parameters: the onsite Hubbard U and the NNN attraction within A and B sublattice V_2 .

Tuning the interacting parameters U and V_2 provides valuable insights into the physical properties of twisted WSe₂, particularly in determining the relevant parameter regime (see Supplemental Material Section II). Here, we provide some further remarks on the values of the interacting parameters. First, the physical properties of the system depend considerably on the Hubbard repulsion U (particularly on U_A , U_B), where smaller U weakens the insulating behavior and larger U stabilizes antiferromagnetic ground state over superconductivity for small displacement field. The Hubbard $U = 35 \sim 40$ meV required to reproduce the experimental results is somewhat weaker than anticipated: this value corresponds to a relatively large dielectric constant $\epsilon \sim 21.4$, estimated by a direct expansion of the dual-gate screened Coulomb interaction using experimentally relevant gate distance $\xi \sim 10$ nm, which can be attributed to the screening effects (e.g., from the high-energy bands) that reduce the interaction strength efficiently. In addition, considering the qualitative nature of the band structure, we may also argue that

Hubbard U could be much larger if the Fermi surface nesting weakens at a small displacement field. Second, although an infinitesimal V_2 can induce superconducting tendency due to Fermi surface instability, a finite $V_2 \sim 10$ meV is still required to better explain the continuous superconductor-insulator transition in the experiment. As previously mentioned, we attribute the continuous transition behavior to disorder, which suggests that in our mean-field calculation without disorder, the system should either exhibit a weak first-order direct superconductor-antiferromagnetic insulator transition, or feature a small intermediate antiferromagnetic metal phase. A weaker V_2 results in extended intermediate phase, while a much larger V_2 will cause strong direct first-order transition, which is challenging to make continuous even when disorder is considered. It is anticipated that a slight modification of the band structure or Hubbard U could lower V_2 considerably.

In summary, while our parameter choices may quantitatively differ from the experimental system, we are confident that, for the reasons outlined above and the supplemental data of tuning parameters, they capture the essential qualitative physics and offer valuable insights into understanding essential physics in the twisted WSe₂ system.

Below please see our point-to-point responses to the reviewers' comments:

Reviewer #1

The authors aimed to provide a mean-field explanation of the recent experimentally observed phase diagram of twisted bilayer WSe₂ (3.65 degrees). The many-body Hamiltonian contains a 3-band (per spin) Wannier model that properly captures the first two bands per spin, an onsite repulsion, and an NNN attraction. With mean-field approximation, the authors obtained a chiral topological superconductor phase at a small displacement field, an AFM insulator at an intermediate displacement field, and an AFM metal at a large magnetic field for hole filling being 1.

The importance of this work relies on its experimental relevance. The obtained superconductor-insulator-metal behavior is consistent with the experiment, which is a huge plus. The authors further provided some predictions about the three phases—nontrivial topology for the superconducting phase and the AFM for the rest, which will certainly motivate future efforts. The only big issue that prevents me from directly recommending the publication of the current version is the reasoning behind the chosen interaction terms. I can imagine the onsite repulsion, but it is hard for me to imagine (and thus easily accept) the NNN attraction, as the authors did not provide any detailed explanation of such a term. The authors mentioned that the NN attraction is less dominant, but why we should choose 10meV for the NNN attraction? What is the logic behind it?

Reply: We sincerely thank the reviewer for valuing the scientific importance of our work. The reasoning behind the inclusion of only onsite Hubbard repulsion and NNN (or NN) attraction in our model is detailed in the “Concern1: Model Terms” section above, based on our physical insights. In summary, we have identified the onsite Hubbard repulsion and NNN attraction as the key interactions for understanding the physics of twisted WSe₂ system, not only for their capability to capture the experimental phenomenology at $\nu = -1$, but also for their close connection to the true microscopic interactions. Focusing on the microscopic aspect, we emphasize that the Coulomb repulsion is short-range due to the strong screening effects of gates (with gate distance comparable to moire lattice constant), justifying the inclusion of only onsite Hubbard repulsion and allowing further neighbor attractions to appear. The phenomenological attraction may arise from various possible microscopic origins (electron-phonon coupling, fluctuations of neighboring insulating phases, or RPA-like overscreening effects, which cannot be definitely identified due to limited experimental data), resulting in significant uncertainties in its form and strength. Given this situation, we refrain from assuming a specific attraction mechanism. Instead, we compare the effects of NN and NNN attraction in the Supplemental Material Section IV, concluding that NNN attraction leads to significantly stronger superconductivity (for the same attraction strength) and identifying the NNN attraction as the key interaction for superconductivity. The logic behind choosing $V_2 \sim 10$ meV for NNN attraction will be addressed in the next reply paragraph (along with the parameter choices) for improved flow.

So I recommend the authors provide more support for the chosen value of the NNN attraction. If the authors can address this request, I would be happy to recommend the publication. Also, it would be good to provide some support for the chosen value of U .

Reply: We appreciate the reviewer’s attention to our parameter choices and the potential recommendation of publication of our work in Nature Communications. We have provided more details of our parameter choices in the “Concern 2: Parameter Choices” section above, as well as in the Supplemental Material Section II and III. By tuning the strength of Hubbard interaction, we estimate that U should take an intermediate strength $35 \sim 40$ meV

to match the insulating behavior observed in the experiment. And we would like to emphasize that, the relatively large value of $V_2 \sim 10$ meV is introduced only for obtaining a relatively weak first order mean-field transition (which can show continuous transition feature with further consideration of disorder) to explain the continuous superconductor-insulator transition observed in the experiment. Actually, superconducting tendency are expected for infinitesimal attraction due to the time-reversal symmetry of the system. Moreover, minor changes to the noninteracting band structure may further reduce the required value of V_2 .

One small issue: the label for the NNN attraction is V in equations 7 and 8, and is changed to V_2 in figure 4. I suggest the authors make the label consistent by sticking to V_2 , since V is already used in equation 2.

Reply: We thank the reviewer for the helpful suggestion. We have changed the notation for the NNN attraction from V to V_2 for consistency and clarity throughout our manuscript.

Reviewer #2

In this work, after constructing a three-band tight-binding model through direct Wannierization and incorporating onsite Hubbard repulsion, the authors theoretically analyzed superconducting and correlated insulator phases within mean-field theory as a displacement field changes at a certain twist angle and filling factor, explaining recent experimental results by Y. Xia et al. in Ref. [71] on twisted bilayer WSe₂. The manuscript demonstrates that for zero or weak displacement fields, a topological superconducting phase with a nontrivial Chern number occurs, while for a relatively strong displacement field, the ground state becomes a correlated antiferromagnetic insulator. The authors argue that the theoretical results are largely consistent with the recent experiments in Ref. [71].

Since mean-field theory typically overestimates a tendency toward broken symmetry states and it is sometimes possible to find a parameter set that exhibits a desired broken symmetry, the proposed mean-field theory needs to be tested with a broader range of parameters for more convincing comparisons with experiments, with a discussion on the validity of the theory. Here are some questions or comments that, in my opinion, need to be clarified in the revised version:

Reply: We thank the reviewer for the thoughtful summary of our work and the recognition of its relevance to experiment results. We are also grateful for the reviewer’s constructive suggestions to help us improve our manuscript.

- The mean-field phase diagrams depend on the detailed choice of interaction parameters, such as the Hubbard repulsions U_A , U_B , and U_C , and next-nearest-neighbor (NNN) attractions V . How can the choice of parameters used in the manuscript be justified? Are the values of these parameters simply chosen to match the observed phases in Ref. [71], or are these parameters estimated independently?

Reply: We appreciate the reviewer’s attention to our parameter choices, which is indeed a crucial point that requires further justification. We have provided more details of our parameter choices in the section “Concern 2: Parameter Choices” above, as well as in the Supplemental Material Section II and III. Though not directly related to this question, you might also find the section “Concern 1: Model Terms” helpful, as it provides a detailed and comprehensive summary of the physical insights underlying our model.

Now we specifically address your question. Some of the parameters are chosen to explain the experimental observations, but they can also be estimated independently. Given that we include only the most important interaction terms in our model and focus on the qualitative physics, treating all interacting parameters as free and tuning them to extract physical insights is a suitable approach for our study. By tuning the strength of Hubbard interaction, we determine that $U = 35 \sim 40$ meV is the appropriate value to match the insulating behavior observed in the experiment. Such Hubbard U corresponds to a relatively large dielectric constant $\epsilon \sim 21.4$ (using dual-gate screened Coulomb interaction with experimentally relevant gate distance $\xi \sim 10$ nm), which can be attributed to the strong screening effects of the twisted WSe₂ system. We also verify that changing U_C does not significantly affect the mean-field phase diagram, as it is away from the Fermi surface. Although an infinitesimal NNN attraction V_2 is sufficient to induce superconducting tendency, we adopt relatively large values of $V_2 \sim 10$ meV to better explain the continuous superconductor-insulator transition observed in the experiment (for details, see the “Concern 2” section above). The attraction terms may originate from various possible microscopic mechanisms, including electron-phonon coupling, fluctuations of neighboring insulating phases, or RPA-like overscreening effects, though their exact nature cannot be definitively determined due to insufficient experimental data so far.

- The authors only considered a filling $\nu = -1$. What is the doping dependence of the superconducting phase? In Ref. [71], different metallic behaviors were observed above and below the filling $\nu = -1$. Does the proposed

mean-field theory reproduce this type of behavior consistent with the experiment?

Reply: We thank the reviewer for this helpful question. The doping dependence of the twisted WSe₂ system is indeed an important aspect that can be further addressed within mean-field theory, and we are happy to provide more discussions on it.

Before presenting the mean-field results, we first offer some general theoretical arguments to understand the doping dependence of twisted WSe₂ near the filling $\nu = -1$ within our mean-field approach. One of the most notable experimental observations is the strong pinning of both the superconducting and insulating phases at $\nu = -1$, which suggests that some commensurate mechanism plays a crucial role in the physical understanding of the 3.65° twisted WSe₂ system. The $\sqrt{3} \times \sqrt{3}$ antiferromagnetic order for the insulating phase is clearly commensurate. Assuming this order persists for doping slightly away from $\nu = -1$, the new filling factor in the enlarged unit cell will deviate from the original integer value $\tilde{\nu} = -3$, leading directly to a metallic phase. In contrast, the superconducting tendency persists (even for infinitesimal attraction) against doping, as the time-reversal symmetry remains intact. Whether doping away from $\nu = -1$ leads to a metal or superconductor depends on the relative energy of these two states.

We provide our mean-field results on the doping dependence of the superconducting phase in the Supplemental Material Section VI. As shown in Fig. S6, the superconducting phase persists only within the doping range $-0.9 \sim 1.2$, while antiferromagnetic (metal) phase becomes energetically favorable when further doping away from $\nu = -1$. Fermi surface analysis indicates that the antiferromagnetic phases on either side of doping have distinct origins: the less hole doped side is closer to the van Hove singularity, resulting in an increased density of states, while the more hole doped side benefits from better nesting conditions for the $\sqrt{3} \times \sqrt{3}$ type order. Thus, the metallic behavior upon doping away from the $\nu = -1$ superconductor can be, at least qualitatively, understood by a mean-field calculation of our model.

- The authors only considered a twist angle of 3.65°. What would be the dependence on the twist angle? In Ref. [72], superconductivity was observed in twisted bilayer WSe₂ at a twist angle of 5°, where correlations may not play a significant role. Is the current theory still applicable to this case, and are the results consistent with the experiment in Ref. [72]? The authors should discuss the results of the experiment in Ref. [72], comparing them with the mean-field calculations.

Reply: We fully agree with the reviewer that examining the applicability of our theory at other twist angles (particularly at 5°, where Ref. [72] provides concrete experimental data) is an important issue, and we appreciate the reviewer's insightful inquiry.

We begin with a brief discussion on the difference between 3.65° and 5° twisted WSe₂ incorporating both theoretical arguments and the experimental phenomenology. First, the key distinction between the two systems is that 3.65° system lies in a more strongly coupled regime compared with 5° system. Theoretically, the kinetic energy scales with twist angle as $E_k \sim \theta^2$, while the interaction energy scales as $E_{int} \sim \theta$. Consequently, the relative ratio of interaction energy to kinetic energy follows $E_{int}/E_k \sim \theta^{-1}$, indicating that smaller twist angle regime is more strongly coupled. Experimentally, the superconducting and insulating phases in the 3.65° system are strongly pinned at $\nu = -1$, whereas these phases in the 5° system are located along the van Hove singularity line, perfectly illustrating the difference in interaction regime. Second, it is theoretically known (see, e.g. [Phys. Rev. Research 6, 033127 (2024)]) that the top moire valence band can be described using a honeycomb model (i.e. A and B sublattices in the main text) when the twist angle is small, while at larger twist angles, it mixes significantly with the C sublattice component. This can be understood by comparing the energy difference δ between the A (or B) and C sublattices induced by moire potential with the typical kinetic energy scale $E_k \sim \theta^2$, where δ tends to exclude the C sublattice from low energy physics, while E_k promotes its mixing.

Considering the above theoretical insights, we conclude that our mean-field theory for the 3.65° twisted WSe₂ can, with some adaption, be applied to the 5° twisted WSe₂. First, we should systematically consider incommensurate orders rather than restricting to commensurate order only, which requires comparing the energies of all possible mean-field ansatz. Second, we should pay more attention to the parameters involving C orbital, as they are expected to vary with the displacement field V_z . As in the footnote [100], the A and B Wannier functions are approximate eigenstates of the displacement field, making these Wannier functions and corresponding parameters approximately independent of V_z . However, this does not apply to the C orbital. In the case of 3.65° twisted WSe₂ discussed in the main text, the C orbital is far from the Fermi surface, justifying the neglect of the V_z dependence of parameters. However, for 5° twisted WSe₂, a more careful analysis of the V_z dependence of parameters is necessary since significant C component is at the Fermi surface. Third, we should retain more long-range parameters, considering the decreasing moiré lattice constant and the increased admixture of the more non-local C orbital. However, it should be kept in mind that, for 5° twisted WSe₂, where the physics is dominated by the van Hove singularity, mean-field calculation is less reliable due to strong fluctuations.

Just as we were considering adapting our mean-field theory to the 5° system, we found a very recent systematic and comprehensive study [arXiv:2412.14296] on that system, which uses a similar three-band tight-binding model obtained through Wannierization of the continuum model. They employ an functional renormalization group approach to obtain a phase diagram consistent with the experimental observations of the 5° system. They conclude that the insulating phase is explained by a generally incommensurate intervalley-coherent antiferromagnetic order, while the superconducting phase has mixed chiral d/p pairing symmetry. They also decrease the twist angle and find that the superconducting order shifts to smaller displacement field and density closer to $\nu = -1$, while still aligning with the van Hove line rather than being pinned at $\nu = -1$. They attribute the mechanism of superconductivity to the antiferromagnetic fluctuations. From our perspective, the results of our 3.65° system and their 5° system share much similarities, such as the superconducting pairing symmetry and the origin of insulating phase, suggesting a similar underlying mechanism. However, directly obtaining phase diagrams at both twist angles using unified model and method remains a challenging task beyond the current ability. This remains an important open question that deserves detailed future studies.

Reviewer #3

The authors report on a mean-field study of WSe₂ homobilayers at $\nu = 1$ filling, aiming to address the nature of the recently observed superconductivity in that system. They describe the low-energy physics of the bilayer by a three-band tight-binding Hamiltonian and obtain the phase diagram as a function of displacement field, V_z , finding a superconducting ground state for small V_z , which transitions into an antiferromagnetic (AFM) insulator and then into an AFM metal as V_z is increased. They conclude that the superconducting phase is topologically non-trivial with inter-valley pairing and mixed $d_{xy} \mp d_{x^2-y^2}$ and $p_x \pm ip_y$ symmetry. The possibility of a topological superconductor in moire TMDs is an interesting and very timely topic. Besides, the numerical results presented in this work seem sound and the manuscript is well written. However, from the way the mean-field Hamiltonian is constructed, it appears that some potentially relevant terms are discarded and it is not clear how the values of the interaction terms used for the calculations are obtained. This suggests that the conclusions of the paper could be only specific to the particular choosing of Hamiltonian an parameters, instead of robust. For this reason, that I elaborate on below, I don't believe the results meet the standards of Nature Communications in terms of potential significance to the field and I recommend the paper to be transferred to a more specific journal, like Communications Physics, unless the authors convince me otherwise.

Reply: We sincerely thank the reviewer for thoughtful evaluation of our work and for recognizing the importance of studying topological superconductivity in moiré TMD systems. We appreciate the positive remarks on the soundness of our numerical results and the clarity of the manuscript presentation. We understand the concern about the Hamiltonian construction and interaction parameters; we would like to provide more clarifications below.

1. The interacting Hamiltonian used in this study only includes on-site Hubbard interactions U_α . On the other hand, it is known that in semiconductor moire materials long-range repulsion terms are non-negligible. Did the authors confirm that adding, for instance, first-neighbor repulsion $+V_1$ does not affect their results? In this sense the calculations seem very biased, as only interacting terms favoring the two phases described are kept.

Reply: We sincerely appreciate the reviewer's attention on our model and insightful comment regarding the potential role of longer-range Coulomb interactions. The reasoning behind the inclusion of only onsite Hubbard repulsion and NNN attraction in our model is detailed in the "Concern1: Model Terms" section above, based on our physical insights. To directly address your question on NN repulsion V_1 , we exclude it in our model due to its relatively small magnitude from screening effects and its negligible impact on the qualitative physics, as detailed below.

First, since the NN Coulomb repulsion V_1 is smaller than the onsite Hubbard U , we exclude it to retain the most relevant interaction terms only. As mentioned in the device fabrication section of [Nature 637, 833 (2025)], the gate distance is comparable to the moiré lattice constant, leading to substantial gate screening effect of Coulomb interaction beyond onsite term. If considering the dual-gate screened Coulomb interaction with experimentally relevant gate distance $\xi \sim 10nm$, we can estimate the ratio of $V_1/U \sim 0.4$ using the Wannier functions (further neighbor interactions are significantly smaller). Although the NN repulsion V_1 may not appear small at first glance, we argue that the effective V_1 should be significantly reduced when considering possible attraction mechanisms in the twisted WSe₂ system (e.g., electron-phonon coupling or antiferromagnetic fluctuations) and potential underestimation of screening effects (e.g., from remote bands). Therefore, considering only onsite Hubbard repulsion should provide a good starting point for qualitatively understanding the physical properties of twisted WSe₂.

Second, as stated in the ‘‘Concern 1’’ section above, longer-range repulsion (e.g. nearest-neighbor repulsion V_1) typically favor CDW order without breaking time-reversal symmetry, making it less relevant in explaining the insulating behavior (due to the spin related degeneracy $E_{\uparrow}(k) = E_{\downarrow}(-k)$). Moreover, such CDW order is unlikely to occur in the twisted WSe₂ system, as the large onsite Hubbard U effectively suppress such ordering tendency. Based on the above considerations, we conclude that CDW order driven by longer-range interactions is largely irrelevant to our study.

Third, an alternative possibility is that the NN repulsion V_1 induces an effective interaction in the crossed particle-hole channel, which energetically favor spin-bond order (SBO). This possibility has been systematically studied and discussed in [arXiv:2412.14296] for 5° twisted WSe₂ using the functional renormalization group technique, with much of the underlying physics strongly related to our 3.65° case. Crucially, both orders transform under the trivial A irreducible representation, thus they are symmetry-allowed to mix without breaking additional symmetries. Including small NN repulsion V_1 would induce finite SBO that mixed with the antiferromagnetic order and only quantitatively shift the mean-field phase boundaries. However, considering the NN repulsion and the corresponding SBO order will not qualitatively alter the phase diagram. Namely, our results are qualitative robust against including an appropriate NN V_1 .

Based on the above analysis, we want to emphasize that the onsite interaction remains the dominant driving mechanism for the insulating behavior in 3.65° twisted WSe₂. Longer-range interaction terms may shift the phase boundaries quantitatively and slightly, but they do not alter the fundamental competition between superconductivity and correlated insulating phase mediated by strong local correlations. Therefore, Our focus on onsite Hubbard U and NNN attraction are sufficient for capturing the main physics in the 3.65° twisted WSe₂.

2. In the manuscript it is not clear where the values for U_{α} and V_2 come from, the choosings seem arbitrary. Are these directly calculated from the Wannier functions? How do we know that these values do not vary significantly with V_z ?

Reply: We are grateful for the reviewer’s focus on our parameter choices, which we agree is a crucial aspect requiring further elaboration. We have provided more details of our parameter choices in the ‘‘Concern 2: Parameter Choices’’ section above, as well as in the Supplemental Material Section II and III.

In response to your question regarding our parameter choices, they are mainly determined by explaining the experimental observations although it is also possible to obtain some of them by direct calculation from the Wannier functions. Considering that we include only the most important interaction terms in our model and focus on the qualitative physics, treating all interacting parameters as free and tuning them to extract physical insights is a suitable approach for our study. We first confirm that modifying U_C does not significantly alter the mean-field phase diagram, since the C orbital is far from the Fermi surface. Consequently, although symmetries only enforce $U_A = U_B$ (while U_C can generally be different), we adopt the simplification $U_A = U_B = U_C = U$ in our model without altering the essential physical properties. By tuning the strength of Hubbard U , we then estimate that U should take an intermediate strength 35 ~ 40 meV to match the insulating behavior observed in the experiment. Note that this choice of U value is also reasonable from direction calculaiton of Wannier functions. Such Hubbard U correspond to dielectric constant $\epsilon \sim 21.4$ (using dual-gate screened Coulomb interaction with experimentally relevant gate distance $\xi \sim 10$ nm), which can be attribute to the strong screening effects of the twisted WSe₂ system. And we emphasize that the relatively large $V_2 \sim 10$ meV is introduced for achieving a relatively weak first order transition in our mean-field results (which can show continuous transition feature in experiments when further considering the effects of disorder) to explain the continuous superconductor-insulator transition observed in the transport experiment. In fact, superconducting tendency is expected even for infinitesimal attraction due to the time-reversal symmetry of the system. Moreover, minor changes to the non-interacting band structure (note its inherent uncertainties) may further reduce required value of V_2 .

We now turn to the dependence of parameters on the displacement field V_z . As noted in the footnote [100], the layer polarized A and B Wannier functions are approximate eigenstates of the displacement field, whereas the layer hybridized C Wannier function is clearly not. Consequently, only A and B Wannier functions remain approximatively invariant under the displacement field, resulting in the approximate invariance of parameters within A and B sublattices. Although parameters involving C sublattice should have some V_z dependence, we can neglect this effect since the C sublattice is far from the Fermi surface (see Fig. 2 in the main text). This approximation simplifies our model by removing displacement field dependence of the model parameters, making our analysis more straightforward and the key physics more transparent.

3. The authors should elaborate more on the mechanism giving rise to an attractive interaction $-V_2$. As it stands, the introduction of the term in Eq. (7) seems arbitrary to favor a particular topological superconducting phase.

Reply: We thank the reviewer for the question regarding the mechanism of the attractive interaction, as it provides further justifications for our study. From our perspective, the attractions may originate microscopically from electron-phonon coupling, magnetic fluctuations of neighboring insulating phases, or RPA-like overscreening effects, which we describe in more detail below.

There are many possible microscopic origins of attraction in twisted WSe_2 . First, similar to conventional BCS superconductors, phonon can play significant role in mediating attractive interaction in twisted WSe_2 . Here, it is worth mentioning another twisted system, twisted bilayer graphene, where substantial evidences suggest that its superconductivity could be phonon mediated. Whether phonon can dominantly contribute to superconductivity in twisted WSe_2 remains an open question that deserves further detailed investigation. Second, it is possible that the attraction arises purely from an electronic mechanism. Since the superconducting phase is adjacent to the antiferromagnetic phase, antiferromagnetic fluctuation can also effectively generate an attraction. More generally, such attraction can be qualitatively captured by RPA calculations. For example, [arXiv:2408.16075] provides RPA calculations for twisted WSe_2 at related twisted angles, demonstrating substantial local attraction (near the NNN regime), which offers a quantitative understanding of the attraction. However, due to limited experimental data so far, we conclude that the attraction mechanism cannot be definitely identified at the current stage.

Given this situation, we would like to emphasize the strength of our study. Rather than assuming a specific attraction mechanism, we compare the effects of NN and NNN attraction in Supplemental Material Section IV, concluding that NNN attraction leads to significantly stronger superconductivity (for the same attraction strength) and identifying the NNN attraction as the key interaction for superconductivity. Moreover, the chiral mixed p/d topological superconductivity has dominant tendency for both NN and NNN attraction, indicating that our result on topological superconductivity remains robust even if realistic interaction contains both NN and NNN components. This can be naturally understood, as stated in the same section of the Supplemental Material, since the chiral superconducting order can fully gap the Fermi surface. On the basis of these considerations, we believe our results are robust regardless of the microscopic origin of the attractive interaction.

4. The authors claim that the superconducting phase they predict “has experimentally observable distinctions from that of the moire Hubbard model”. The manuscript falls short in describing these distinctions. They should do an effort to guide experimentalists on how to confirm the phase they are predicting.

Reply: We appreciate the reviewer’s attention to the comparison between our model and the extensively studied (triangular lattice single-band) moire Hubbard model. This is an excellent question that involves some subtle but important aspects of modeling the twisted WSe_2 system. We shall elaborate important distinctions below. We also added more discussion on it in our revised manuscript.

As stated in the main text, the key differences between our model and the moire Hubbard model are mainly twofold. First, our model correctly captures the nontrivial band topology (for the top two moire valence bands) of the twisted WSe_2 , in agreement with the continuum model description; but the single-band moire Hubbard model cannot have a nontrivial band topology due to Wannier obstructions. Second, in contrast to the moire Hubbard model, our model has no emergent spin-valley $\text{SU}(2)$ symmetry at zero displacement field. We would like to emphasize that the emergent spin-valley $\text{SU}(2)$ symmetry in the moiré Hubbard model is actually an artifact of using single-band description. More specifically, as is evident from the continuum model description, the twisted WSe_2 system has time-reversal symmetry and pseudo-inversion symmetry (when considering the lowest-order harmonic terms only). In a single-band model (like the moire Hubbard model), these two symmetries constrain all hopping parameters to be real and enforce identical spin up and spin down hoppings, directly leading to the emergent spin-valley $\text{SU}(2)$ symmetry. However, for a multiband system, the above argument no longer holds due to the presence of additional orbital degrees of freedom. Two additional remarks: First, $\text{SU}(2)$ symmetry is not required to ensure the spin degeneracy of the band, time-reversal and pseudo-inversion symmetry alone are sufficient. Second, $\text{SU}(2)$ symmetry is absent microscopically because of the strong Ising-type spin-orbit coupling in TMD systems.

The actual absence of spin-valley $\text{SU}(2)$ symmetry, as opposed to its presence, has striking physical consequences that are experimentally observable. The key distinction between the two models is the superconducting pairing symmetry they predict. In our model, the absence of spin-valley $\text{SU}(2)$ symmetry allows singlet and triplet pairing to mix, which manifests in our predicted mixing of chiral $d \pm id$ and $p \mp ip$ pairing symmetry. However, the presence of spin-valley $\text{SU}(2)$ symmetry forbids singlet and triplet pairing from mixing, thereby preventing any hybridization between d-wave and p-wave components. Experimental techniques capable of detecting the superconducting pairing symmetry (e.g. phase sensitive measurements or spectroscopic measurements) should in principle reveal this distinction. Moreover, though more difficult for experimental detection, topological superconductivity is classified differently for the presence of $\text{SU}(2)$ spin-valley symmetry compared with the current $\text{U}(1)$ symmetry case. These distinctions are, in our view, fundamental for advancing experimental studies of superconductivity in twisted WSe_2 .

5. In the large V_z regime of the phase diagram in Fig 4(a), it seems that the SC re-enters – the dashed blue line

has lower ΔE than the green line. Can the authors explain this?

Reply: We sincerely thank the reviewer for pointing this out, which is invaluable for improving our manuscript. Indeed, in the previous version of our manuscript, the superconducting phase slightly reenters at relatively large displacement field V_z when the NNN attraction is increased to $V_2 = 12.5$ meV. This increase in V_2 was intended for a direct phase transition between superconducting and antiferromagnetic insulating phase (to account for the continuous superconductor-insulator transition, see the ‘‘Concern 1’’ section and the response of your second question above), but led to a slight reemergence of superconductivity. However, we affirm that the key conclusions of our manuscript remain intact. The most important insight from our mean-field results in the large V_z regime is that both the superconducting and antiferromagnetic order are strongly suppressed due to the decrease in density of states, which is consistent with the experimental observations. The competition between the superconducting and antiferromagnetic metal phases at large V_z depends on the details of the parameter choices, and the reemergence of superconductivity is not a robust feature and can be avoided. To address this issue, we adopt slightly stronger Hubbard repulsion parameters $U_A = U_B = U_C = 37.5$ meV in our revised manuscript such that the enhanced antiferromagnetic order can prevent the slight reentrance of superconductivity in the large V_z regime.

6. Regarding footnote [100], the fact that the topmost moire band has almost no ‘‘C-component’’ is only true for a particular regime of twist angles depending on the continuum model parameters that one chooses. I’m not sure if this approximation is valid in general.

Reply: We fully agree with the reviewer that whether the topmost moire band has almost no C component depends on specific conditions of twist angle and continuum model parameters. More precisely, this statement holds true in the relatively small twist angle regime for certain twisted TMD homobilayers (including twisted WSe₂). The work [Phys. Rev. Research 6, 033127 (2024)] provides a clear physical picture of the crossover in low-energy orbitals, from predominantly honeycomb orbitals (i.e. A and B sublattices in the main text) in the small twist angle regime to significant mixing with the C orbital at larger twist angles, which we briefly explain below. We start by emphasizing two important energy scales in twisted TMD homobilayers that determines the low energy behaviors of the Wannier orbitals. The first one is the moire potential difference δ between A (or B) and C sublattice, which have only relatively weak twist angle dependence. Whether A/B sublattice or C sublattice has lower energy depends on the details of the continuum model parameters (see e.g. [Phys. Rev. Research 2, 033087 (2020)]). In our case of twisted WSe₂, A and B sublattices have lower energy, favoring a honeycomb low energy description. The second one is the characteristic kinetic energy scale, which varies with the twist angle as $E_k \sim \theta^2$, indicating that a larger twist angle results in substantially higher hopping energy. In the small twist angle regime, the moire potential effect δ dominates (i.e. only A and B sublattices are at low energy), resulting in the topmost moire band having almost no C component. As the twist angle increases, the hopping energy becomes more significant, making the effect of moire potential δ less prominent and enhancing the mixing of the C component in the topmost moire band. According to our Wannierization results for 3.65° twisted WSe₂, the system falls within the small to intermediate twist angle regime, where only a relatively small C component is near the Fermi surface (see Fig. 2 in the main text). We would also like to emphasize that this footnote, regarding the topmost moire band having almost no C component, is written to justify neglecting the displacement field V_z dependence of the model parameters, as the parameters involving C orbital are expected to be V_z dependent. For further details, you may refer to our response to your second question on the V_z dependence of the parameters.

7. Should we expect that corrections to the Hamiltonian that are beyond the mean-field approximation do not destabilize the topological superconducting ground state? If so, why?

Reply: We appreciate the reviewer’s question regarding the stability of the topological superconducting ground state beyond the mean-field approximation. In response to your question, the topological superconducting phase is expected to remain robust against fluctuations beyond the mean-field approximation, as it originates from the Cooper instability of the Fermi surface. As long as the system exhibits some attractive interactions (for possible mechanisms, see our response to your third question), the perfect nesting in the particle-particle channel strongly favors superconducting phase, which is unlikely to be destabilized by fluctuation effects. We anticipate that many-body numerical techniques beyond the mean-field approximation can be used to further confirm the topological superconducting phase in future studies.

8. It’s hard to read the insets of Fig 4 in the printed version. Could the authors increase their size?

Reply: We sincerely thank the reviewer for pointing this out. We have enlarged Fig. 4 in our revised manuscript to improve its readability in the printed version.

As a final point, we would like to express our sincere gratitude once again to all three reviewers for their valuable and insightful comments. We hope that the revised manuscript and our detailed point-to-point responses meet your expectations. We remain happy to address any additional concerns or suggestions to further improve the quality of our work.

Response to Reviewers' Comments for Manuscript NCOMMS-24-59815A

Chuyi Tuo,^{1,*} Ming-Rui Li,^{1,2,*} Zhengzhi Wu,^{1,3} Wen Sun,¹ and Hong Yao^{1,†}

¹*Institute for Advanced Study, Tsinghua University, Beijing 100084, China*

²*Department of Physics, Princeton University, Princeton, New Jersey 08544, USA*

³*Rudolf Peierls Centre for Theoretical Physics, Parks Road, Oxford, OX1 3PU, UK*

We sincerely thank both reviewers for their insightful and constructive feedback, which has been invaluable in enhancing the quality and clarity of our manuscript. We have thoroughly revised the manuscript based on the reviewers' comments and suggestions.

The key revisions made to the manuscript are summarized as follows:

1. *In response to the comments of Reviewer #1, we added one additional section to the Supplemental Material to provide further microscopic justification for the attractive interaction:*
“V. Attractive Interactions from Random Phase Approximation Analysis”
2. *In response to the comments of Reviewer #3, we added one additional section to the Supplemental Material to discuss the displacement field dependence of the Wannier functions:*
“II. Displacement Field Dependence of the Wannier Functions”

Below please see our point-to-point responses to the reviewers' comments:

Reviewer #1

Unfortunately, I am not convinced by the authors' justification for the next-nearest-neighbor (NNN) attractive interaction. After reviewing their response, I find no independent reasoning for selecting this interaction other than numerically reproducing the experimentally observed phases. Given this, I have to recommend that the manuscript be published in a more specialized journal rather than Nature Communications.

Reply: Thank you very much for your time further reviewing our manuscript and for your helpful feedback. We understand that an independent theoretical justification for the attractive interaction terms in our model was a concern, and we have undertaken a substantial revision to address it.

In the revised manuscript, we have performed detailed theoretical analysis based on the random phase approximation (RPA) to provide a microscopic justification for the attractive interactions adopted in our model, as detailed in Sec. V of the Supplemental Material. The RPA method captures collective screening effects by summing the infinite series of bubble diagrams in the particle-hole channel, allowing for a nontrivial renormalization of interactions that can include effective attraction in specific real-space regions. In the Supplemental Material, we carefully analyze the RPA screening effects of dual-gate screened Coulomb interaction projected to the topmost moiré band, where the resulting RPA effective interaction is demonstrated in Fig. S5. We first observe that the RPA mechanism further screens the Coulomb interaction, resulting in enhanced localization of the repulsive component, which further supports the use of only onsite Hubbard interactions in our model. More importantly, the RPA analysis reveals that substantial attractive interactions can emerge predominantly within relatively local regions ($r \lesssim a_M$), thereby justifying the inclusion of local NNN/NN attraction terms in our tight-binding framework. We mainly focus on the NNN attraction in the main text, as it leads to significantly stronger superconductivity (see Fig. S6), which can be understood from the Fermi surface analysis discussed therein.

We believe that the inclusion of the RPA analysis directly addresses your main concern and substantially strengthens the theoretical foundation of our work. It provides microscopic support for the choice of interaction terms, moving beyond purely phenomenological reasoning.

* These two authors contribute equally in this work

† yaohong@tsinghua.edu.cn

Reviewer #3

The field of superconductivity in semiconductor moiré materials is at a very early stage, as there are only two experimental reports up to date. Besides, the model parameters that enter in the continuum descriptions of WSe₂ are obtained from DFT, which may have high uncertainties and depend on the functionals used. In this context, the approach taken from the authors to isolate the key physics rather than incorporating the full microscopic details is justified. The additional sections added to the supplemental material help supporting this point, as they show that the values taken for the interaction terms qualitatively reproduce several aspects of the experimental phenomenology. In the previous version of the manuscript the values taken for the mean-field Hamiltonian parameters were lacking a better justification. I recommend that the authors add a short comment in the introduction about the approach taken, something along the lines of what they comment in the reply: that they expect their choices to capture the essential qualitative physics of the system.

Reply: We thank Reviewer #3 for his/her thoughtful and encouraging feedback, and appreciate his/her positive recognition that our approach of isolating the key physics is justified. Following your suggestion, we have added a short comment in the introduction section to clarify that our model aims to capture the essential qualitative physics. We appreciate the reviewer's suggestion, which has helped us improve the clarity of our work.

The authors have address all of my concerns. Because they are focusing on $\nu=1$, I agree that it is justified to just keep on-site interactions to capture the main physics and I also think it's okay to leave an explanation of the microscopic origin of attractive interactions for future work.

Reply: We sincerely thank Reviewer #3 for his/her encouraging comments. We are glad that the focus on onsite interactions at $\nu = -1$ is seen as appropriate, and appreciate your understanding that a full microscopic explanation of the attractive interaction can reasonably be left for future work. To further strengthen the theoretical basis of our model, we have now included an RPA analysis in Sec. V of the Supplemental Material to provide a more microscopic justification for the attractive interaction terms.

I only have one final question: Have the authors numerically checked that the top band has no C-orbital component in the large V_z limit? How independent of V_z are the A and B Wannier functions really?

Reply: We appreciate the reviewer's thoughtful question. We have now explicitly examined the displacement field dependence of the Wannier functions, and the results are summarized in Sec. II of the Supplemental Material. Specifically, we explicitly perform the same Wannierization procedure for the continuum model with displacement field set to relatively large value $V_z = 20$ meV, and the resulting Wannier functions are shown in Fig. S2. Compared with the Wannier functions at zero displacement field (Fig. 1(b)), we find that the A and B Wannier functions remain almost unchanged while the C Wannier function shows noticeable redistribution, which is consistent with the discussion in Footnote [105]. In the main text, we neglect this modification of the C orbital since it lies far from the Fermi surface and does not significantly alter the low-energy physics. To further justify this approximation, we compute the onsite chemical potential terms directly from the Wannier functions at $V_z = 20$ meV: $t_0^A = -7.77$ meV, $t_0^B = -22.72$ meV, and $t_0^C = -41.10$ meV. These values confirm that the C orbital remains energetically well separated from the low-energy physics.

Regarding the reply to Reviewer 2: I think the authors provide satisfactory answers. I would suggest to add a comment in the discussion section about the similarities and differences of the 3.65 degree system that they study with respect to the one at 5 degree twist (mainly summarizing what they discuss on the reply.) I believe this would improve the quality of the paper.

Reply: We thank the reviewer for this suggestion. We have added a brief comment on the 5° tWSe₂ system in the discussion section and cited the relevant reference to guide interested readers. We agree that this addition improves the completeness and clarity of the manuscript.

For the previous reasons I recommend this work to be published in Nature Communications.

Reply: We sincerely thank Reviewer #3 for his/her positive evaluation and recommendation for publication in Nature Communications. We greatly appreciate your thoughtful comments and support throughout the review process.

In summary, we would like to thank all reviewers again for their careful evaluation and constructive feedback. We hope that the revised manuscript is now suitable for publication in Nature Communications and remain happy to clarify any further points.

Response to Reviewers' Comments for Manuscript NCOMMS-24-59815B-Z

Chuyi Tuo,^{1,*} Ming-Rui Li,^{1,2,*} Zhengzhi Wu,^{1,3} Wen Sun,¹ and Hong Yao^{1,†}

¹*Institute for Advanced Study, Tsinghua University, Beijing 100084, China*

²*Department of Physics, Princeton University, Princeton, New Jersey 08544, USA*

³*Rudolf Peierls Centre for Theoretical Physics, Parks Road, Oxford, OX1 3PU, UK*

We sincerely thank all reviewers for their thoughtful and constructive feedback. Their comments have significantly improved the clarity, rigor, and overall quality of our work. We are also grateful to the editor for the careful handling of our submission throughout the review process.

Below please see our point-to-point responses to the reviewers' comments:

Reviewer #3

The authors have satisfactorily addressed my additional question. I recommend this work to be published in Nature Communications.

Reply: We thank the reviewer for their positive evaluation and recommendation for publication. We are pleased that our revisions have satisfactorily addressed the reviewer's concerns.

Reviewer #4

Reviewer #1's primary concern about this manuscript pertained to the justification of the next-nearest-neighbor attractive interaction in the model. This attractive interaction is essential for the emergence of superconductivity, according to the authors' analysis.

In the resubmitted supplementary material, the authors present an RPA analysis for the renormalization of interactions. While this RPA analysis leads to a weaker attractive interaction strength compared to the values of V_2 adopted in the main text, it does demonstrate a microscopic mechanism for the appearance of an attractive interaction at a finite range in the twisted WSe2 system. In my view, the reviewer #1's concern has been largely addressed.

Reply: We sincerely thank the reviewer for the careful reading of our work and the constructive comments. The justification of the NNN attraction is indeed essential to our study. Our RPA analysis demonstrates that a clear effective attraction emerges primarily within NN and NNN ranges. While the strength of the attraction is quantitatively weaker than the phenomenological value adopted, this calculation remains highly valuable. As the reviewer emphasized, it provides a microscopic mechanism for the origin of attraction in twisted WSe2. Moreover, by revealing the spatial structure of the interaction, it further supports the form of the Hamiltonian used in our study. We are grateful to the reviewer for highlighting these important aspects and for recognizing that our RPA calculation has largely resolved Reviewer #1's concern.

* These two authors contribute equally in this work

† yaohong@tsinghua.edu.cn

Review for “Theory of Topological Superconductivity and Antiferromagnetic Correlated Insulators in Twisted Bilayer WSe₂”

The authors report on a mean-field study of WSe₂ homobilayers at $\nu = 1$ -filling, aiming to address the nature of the recently observed superconductivity in that system. They describe the low-energy physics of the bilayer by a three-band tight-binding Hamiltonian and obtain the phase diagram as a function of displacement field, V_z , finding a superconducting ground state for small V_z , which transitions into an antiferromagnetic (AFM) insulator and then into an AFM metal as V_z is increased. They conclude that the superconducting phase is topologically non-trivial with inter-valley pairing and mixed $d_{xy} \mp d_{x^2-y^2}$ and $p_x \pm ip_y$ symmetry.

The possibility of a topological superconductor in moiré TMDs is an interesting and very timely topic. Besides, the numerical results presented in this work seem sound and the manuscript is well written. However, from the way the mean-field Hamiltonian is constructed, it appears that some potentially relevant terms are discarded and it is not clear how the values of the interaction terms used for the calculations are obtained. This suggests that the conclusions of the paper could be only specific to the particular choosing of Hamiltonian an parameters, instead of robust.

For this reason, that I elaborate on below, I don't believe the results meet the standards of Nature Communications in terms of potential significance to the field and I recommend the paper to be transferred to a more specific journal, like Communications Physics, unless the authors convince me otherwise.

1. The interacting Hamiltonian used in this study only includes on-site Hubbard interactions U_α . On the other hand, it is known that in semiconductor moiré materials long-range repulsion terms are non-negligible. Did the authors confirm that adding, for instance, first-neighbor repulsion $+V_1$ does not affect their results? In this sense the calculations seem very biased, as only interacting terms favoring the two phases described are kept.

2. In the manuscript it is not clear where the values for U_α and V_2 come from, the choosings seem arbitrary. Are these directly calculated from the Wannier functions? How do we know that these values do not vary significantly with V_z ?
3. The authors should elaborate more on the mechanism giving rise to an attractive interaction $-V_2$. As it stands, the introduction of the term in Eq. (7) seems arbitrary to favor a particular topological superconducting phase.
4. The authors claim that the superconducting phase they predict “has experimentally observable distinctions from that of the moire Hubbard model”. The manuscript falls short in describing these distinctions. They should do an effort to guide experimentalists on how to confirm the phase they are predicting.
5. In the large V_z regime of the phase diagram in Fig 4(a), it seems that the SC re-enters – the dashed blue line has lower ΔE than the green line. Can the authors explain this?
6. Regarding footnote [100], the fact that the topmost moiré band has almost no “C-component” is only true for a particular regime of twist angles depending on the continuum model parameters that one chooses. I’m not sure if this approximation is valid in general.
7. Should we expect that corrections to the Hamiltonian that are beyond the mean-field approximation do not destabilize the topological superconducting ground state? If so, why?
8. It’s hard to read the insets of Fig 4 in the printed version. Could the authors increase their size?